# Pain Modulation in Chronic Musculoskeletal Disorders: Botulinum Toxin, a Descriptive Analysis

**DOI:** 10.3390/biomedicines11071888

**Published:** 2023-07-03

**Authors:** Daniela Poenaru, Miruna Ioana Sandulescu, Delia Cinteza

**Affiliations:** 1Rehabilitation Department 1, Carol Davila University of Medicine and Pharmacy, 4192910 Bucharest, Romania; delia.cinteza@umfcd.ro; 2Doctoral School, Carol Davila University of Medicine and Pharmacy, 4192910 Bucharest, Romania

**Keywords:** botulinum neurotoxin, neuropathic pain, musculoskeletal disorders

## Abstract

Botulinum neurotoxin (BoNT), a product of Clostridium botulinum, reversibly inhibits the presynaptic release of the neurotransmitter acetylcholine at the neuromuscular junction. In addition, BoNT blocks the transmission of other substances involved in pain perception and, together with a soft-tissue anti-inflammatory effect, may play a role in analgesia. When first-line treatment fails, second-line therapies might include BoNT. Studies on chronic and recurrent pain using different mechanisms offer heterogenous results that must be validated and standardized. Plantar fasciitis, severe knee osteoarthritis, painful knee and hip arthroplasty, antalgic muscular contractures, and neuropathic and myofascial pain syndromes may benefit from the administration of BoNT. Research on this topic has revealed the main musculoskeletal conditions that can benefit from BoNT, stressing the effects, modalities of administration, doses, and schedule.

## 1. Introduction

Botulinum neurotoxin (BoNT) is a product of Clostridium botulinum. BoNT acts on specific proteins, SNARE (soluble N-ethylmaleimide-sensitive factor attachment protein receptors), which mediate neuronal exocytosis and reversibly inhibit the presynaptic release of neurotransmitters at the neuromuscular junction, with acetylcholine and substance P (a neuropeptide acting as a neurotransmitter and a modulator of pain perception) being primarily affected. Acetylcholine reduction causes muscular weakness and substance P reduction is responsible for analgesia [1].

There are seven BoNT serotypes, termed A to G. Each serotype cleaves one of the SNARE proteins (soluble N-ethylmaleimide-sensitive factor attachment protein receptors). BoNT serotypes A and B have been approved for pharmacological use. BoNT/A has three formulations: onabotulinumtoxin A (Botox, Allergan), abobotulinumtoxin A (Dysport, Medicis), and incobotulinumtoxin A (Xeomin, Merz). Serotype B is available under the name rimabotulinumtoxin B (Myobloc, Neurobloc) [2]. The equivalence between BoNT/A and BoNT/B ranges from 1:40 to 1:67 UI.

Generally, the main indications of BoNT are focal dystonia, spasticity (upper motor neuron lesion), non-dystonic disorders of involuntary muscle activity, strabismus, smooth muscle hyperactive disorders, sweating and salivary disorders, and cosmetic use.

Chronic pain syndromes with recurrent or refractory courses are difficult to manage and are the subjects of extensive research. BoNT may have a place in the treatment of some diseases, mainly when more conventional therapies fail (See Table 1, Figure 1).

## 2. Relevant Literature

The aim of the study was to underline some of the painful musculoskeletal conditions, which, in the relapsing or refractory course, may be treated using different regimens of BoNT.

We searched PubMed, Embase, and Cochrane databases for articles published since 2000 using the MeSH terms: botulinum toxin, pain, and musculoskeletal, and excluded all records containing the terms spasticity and dystonia. We collected 52 articles matching the selection criteria (See Table 2). We grouped the articles based on the topics below:

### 2.1. Plantar Fasciitis

Plantar fasciitis often has a chronic evolution, with recurrences or treatment refractoriness. The last therapeutic resort is surgery, with debridement and release of fascia. Trying to avoid the shortcomings of surgery has raised interest in new therapeutic agents such as BoNT [55].

Heel pain is usually caused by plantar fasciitis, although this may not always be the case. Neuropathic conditions such as entrapment of the posterior tibialis nerve, inferior calcaneal nerve, medial plantar nerve, and lateral plantar nerve may mimic or be associated with plantar fasciitis. Quadratus plantae and abductor hallucis brevis muscles exert pressure on and entrap the above-mentioned nerves [56].

Studies in the literature used different pathways for botulinum toxin injections: directly into the fascia, into the short plantar muscles, or in the gastrocnemius-soleus complex.

Intrafascial injection techniques vary based on different studies. Babcock used two doses of intrafascial injection of 70 UI BoNT/A (Botox; Allergan, Irvine, CA, USA) under palpatory guidance: 40 UI in the tender region of the heel medial to the insertion of fascia and 30 UI in the tender point of the arch of the foot (one inch distal to the talar insertion of fascia) [8]. Using this technique, two articles reported significant improvement in pain and function, both in the short-term (3 and 8 weeks) and the long-term (6 and 12 months) [3,8]. Plantar fascia thickness decreased on ultrasound examination at 1 and 3 weeks and was correlated with clinical improvement [6]. Comparing this regimen with local corticosteroids and anaesthetics showed that both treatments reduced pain at one month; however, botulinum toxin was superior at 6 and 12 months, with a remanent effect long after termination of direct pharmacological action [4]. Another injection technique used a dose of 200 UI BoNT/A (Dysport, Ipsen Pharma, Ettlingen, Germany), equivalent to 30–40 UI Botox), administered into the most tender point via plantar approach and into the subfascial area in four different directions, producing pain relief and pressure pain threshold reduction at 2 weeks that was maintained for up to 52 weeks. However, researchers found no significant difference when it was compared with the placebo [4,7]. Ultrasound-guided injection of 50 UI botulinum toxin A into the fascia with a posterior approach below the calcaneus reduced pain at three months. Additionally, the ultrasound also validated a reduction in fascia thickness [9].

Intramuscular injections were administered into the flexor digitorum brevis, abductor hallucis, quadratus plantae, and medial gastrocnemius muscles.

Intramuscular injection of 100 UI incobotulinumtoxin A (Xeomin, Merz) into the flexor digitorum brevis muscle in the proximity of plantar fascia with electromyographic validation produced significant pain and reduced dysfunction [10]. Intramuscular injection (abductor hallucis brevis, 50 UI; quadratus plantae, 50 UI Dysport) under EMG guidance with subsequent paralysis of these muscles and decompression of nervous branches relieved pain and improved function during daily activities. Supplementary analgesia may result from the toxin’s diffusion to the plantar medial heel, with direct neuron-analgesic and musculoskeletal anti-inflammatory effects [11].

Recent studies suggest that gastrocnemius muscle tightness (medial head) and ankle dorsiflexion restriction play an important role in chronic plantar fasciitis [57]. Based on this assumption, researchers injected 70 UI botulinum toxin A into the medial head of the gastrocnemius. Afterward, patients attempted a therapeutic program with conservative modalities (physiotherapy). There was a significant functional and subjective improvement compared with the placebo at 6 and 12 months [12].

Some researchers combined the two paradigms—intrafascial and intramuscular administration—to treat refractory or relapse cases [8].

None of the studies mentioned above reported adverse reactions.

BoNT may be considered an alternative for recurrent or refractory plantar fasciitis, with local, intrafascial administration of small doses or intramuscular injection in the small plantar flexor muscles or calf muscles.

### 2.2. Knee Osteoarthritis

There is currently no cure or ability to reverse the degenerative process; the main objective of any treatment should be alleviating pain, decreasing inflammation, restoring function, and decelerating the progression of the disease. Pain modulation for maintaining function when intra-articular corticosteroids and other first- and second-line therapies have failed or are not indicated may benefit from BoNT administration. Its analgesic and anti-inflammatory properties are due to the inhibition of the release of some neurotransmitters from nerve terminals, including substance P, calcitonin gene-related peptide (CGRP), and neurokinin A [58].

The first attempts to treat knee osteoarthritis using intra-articular botulinum toxin were made in 2010 when Boon et al. reported that 100 UI BoNT/A together with corticosteroids significantly improved knee pain and function. Since then, research has continued to generate evidence to support the use of botulinum toxin [13].

Research on treating different grades of knee osteoarthritis (II, III, and IV Kelgren-Lawrence stage [58]) with intraarticular injections of 100 UI BoNT/A (Botox) or 250 UI BoNT/A (Dysport) reported pain and functional improvement lasting from one week to six months. A second identical injection administered after three months may act as a booster, leading to the assumption that the therapeutic result is transient rather than long-term [14,15,16].

The results for moderate chronic knee osteoarthritis are heterogeneous. Comparing intra-articular doses of 100 UI BoNT/A and 200 UI BoNT/A combined with corticosteroids decreased pain in every group at 8 weeks, with even low-dose BoNT/A reaching statistical significance [59]. In another study, 100 UI BoNT/A had no effect on pain and function, either in the short-term or the medium-term [17].

Intra-articular botulinum toxin was safe, with no adverse effects. In intramuscular administration, adverse reactions were rare and, when they occurred, were mild and transient, including local pain and clinical signs of toxin diffusion into the adjacent tissue (dysphagia following neck muscle injection; diplopia or ptosis following orbicularis oculi injection; skin rash, dizziness, generalized fatigue, dry mouth, reduced sweating, and constipation). Studies on intra-articular administration revealed none of the above-mentioned reactions [14,15,60].

BoNT/A and hyaluronic acid were both efficient at reducing pain and disability and improving activities of daily living when followed by therapeutic exercise, with better scores for BoNT/A in the short-term (4 weeks) and medium-term (8 weeks) [18].

Pain and dysfunction due to chronic knee osteoarthritis may improve after one or two intra-articular injections of BoNT/A, although the literature on this is scarce. It is important to note that there are a multitude of intra-articular preparations and choosing between them is challenging.

### 2.3. Painful Knee Arthroplasty

Painful knee arthroplasty is a challenge, since after the exclusion of infection, loosening, and instability, medical or surgical alternatives are scarce. In this situation, intra-articular administration of 100 UI BoNT/A (Botox) reduced pain and improved function after 2 months, strengthening its short-term analgesic effect. With an average of 39 days of pain relief, there was a need for a second injection or a higher dose to sustain the analgesic effect for a longer duration [19]. The researchers administered a second and third dose of botulinum toxin (100–300 UI) in cases where pain recurred or responses failed.

The results encouraged the use of botulinum toxin for refractory, chronically painful total joint arthroplasty not amenable to medical and surgical treatment [19].

### 2.4. Joint Contractures

Postoperative knee flexion contracture occurs in 15–20% of patients and has important biomechanical consequences. Most studies recommend botulinum toxin for muscle spasticity, i.e., upper motor neuron lesions. In the case of knee arthroplasty, hamstring tightness is secondary to local causes. After palpation of the most tender point of the medial and lateral hamstring muscle bellies, 50 UI botulinum toxin was injected into each point and the patient was assigned a program of physical therapy. One month later, the knee’s range of motion improved and results were maintained for one year. The injection reduced muscular tightness and allowed the patient to attempt rehabilitation to achieve the correct range of motion [20].

Moderate-to-severe flexion contractures (greater than 15°) require surgical intervention—one of the many therapeutic procedures conducted under anaesthesia. Peroneal nerve decompression, posterior soft tissue release (capsule and tendons), and botulinum toxin injection (150–200 UI) into the gastrocnemius and hamstrings can be efficacious treatments with complete correction of the deformity. Botulinum toxin has been proven to be a valuable tool for improving the alignment and consequent disability [21].

Knee flexion contracture is a frequent complication in children and young adult haemophilic patients. Acute joint distension and pain block quadriceps function and create a muscular imbalance in favour of knee flexors. Muscle haematoma and advanced knee arthropathy are associated with flexor contracture due to limited mobility or severe internal joint damage [23]. When a 6-month rehabilitation program fails to restore knee extension, intramuscular botulinum toxin administration may be considered, especially for knee contractures between 10° and 30°, usually secondary to hemarthrosis with inadequate replacement therapy. For knee contractures ranging from 31° to 45° and with muscle imbalance, intramuscular toxin administration may precede hamstring release. There were no benefits for flexion contractures above 45°. A total dose of 100 UI BoNT/A was injected into the hamstrings, fascia lata, and calf muscles under factor replacement protection [23].

In the same manner, a small proportion of patients with total hip arthroplasty develop hip contractures, limiting the range of motion and, ultimately, the gait and quality of life. The most frequently involved muscles are adductors, tensor fascia lata, and rectus femoris [61]. Various attempts have been made to limit these contractures—including surgical techniques associated with rehabilitation programs—with the rate of success varying between 0 and 60% [62]. Researchers proposed intramuscular administration of 100 UI into the adductor magnus and brevis, rectus femoris, and tensor fascia lata, based on clinical examination. After this procedure, the patients started a rehabilitation regimen. At the one-year follow-up visit, gain of hip mobility was on average 23° for those with restricted movement, with full extension to standing [24].

Thus, muscular contractures may benefit from locally administered BoNT/A once a thorough clinical examination has determined the specific muscle to inject.

### 2.5. Chronic Lateral Epicondylitis

Chronic lateral epicondylitis, a degenerative inflammatory disorder, may follow a chronic or recurrent course. The rationale behind recommending botulinum toxin is supported by two mechanisms: a paralysing effect on extensor muscles that reduces tension in their tendons (protection) and an analgesic effect due to inhibition of the neurotransmitters involved in pain transmission [1].

BoNT/A, tested in doses varying from 40 to 60 UI via unique injections, provided significant improvement in visual analogue scale (VAS) pain assessment at week 4 that persisted to week 12 and led to no or minimal alteration in grip strength. The administration site varied across studies (intratendinous or intramuscular). Intratendinous administration site was 1 cm distal to the lateral epicondyle toward the tender spot [25]. Intramuscular administration varied between studies: 3 to 4 cm distal to the lateral epicondyle with infiltration of the muscle at two locations [26], 5 cm distal to the area of maximal tenderness at the lateral epicondyle, in line with the middle of the wrist [27], or at the site where the motor nerve branch (posterior interosseous nerve) enters the extensor digitorum and extensor carpi ulnaris muscles. This point is at a distance of 33% of the forearm length from the lateral epicondyle to the posterior midpoint of the wrist [28]. This last technique led to the largest reduction in pain at rest and of pain at maximum pinch, with the appearance of a transitory extensor lag that interfered with work activities and disappeared at week 16. Thus, it might be useful for patients for whom the development of extensor lag does not affect their ability to work.

Intramuscular administration of low doses of botulinum toxin into the extensor carpi radialis brevis muscle under electromyographic guidance was confirmed to have an analgesic effect at 30 and 90 days and spared the extensor digitorum muscle, avoiding the extensor lag with no interference to daily and work activities. Thus, it seems that targeting the right muscle using a low dose may induce short-term pain reduction and functional improvement [29]. Long-term evaluation found that 40% of patients asked for a second injection at 6 or 9 months. At one year follow-up, patients receiving one or two doses of botulinum toxin reported successful and sustained pain relief, with no need for another therapeutic intervention [30].

The role of botulinum toxin in the therapeutic armamentarium of lateral epicondylitis has to be defined, as researchers have documented an equivalent effect, in terms of pain relief, using corticosteroids. It is worth mentioning that corticosteroids provide analgesia in the early stages while the analgesic effect of botulinum toxin could last for 16 weeks. Recurrence rate and risks of corticosteroid injections are important issues to consider [31]. On the other hand, adverse effects of botulinum toxin, including local paresthesia, ecchymosis, and skin irritation, need to be mentioned. The high costs of this product should also be considered [63].

In the case of chronic refractory lateral epicondylitis, the decision between local corticosteroids and BoNT/A may be inclined towards the latter, considering the resolution of pain and dysfunction and the low incidence of adverse effects.

### 2.6. Neuropathic Pain

Focal postherpetic and post-traumatic pain, diabetic neuropathy, complex regional pain syndrome, and trigeminal neuralgia are subjects of scientific research.

Focal postherpetic and post-traumatic pain with mapping of the precise area of allodynia and intradermal injection of 200 UI BoNT/A (Botox) following the hyperhidrosis scheme were followed by a reduction in the intensity and the area of pain, as well as a decrease in cold and pain thresholds, although perception thresholds remained constant [64]. First-line trigeminal neuralgia treatment is represented by carbamezepine and oxcarbazepine, with good efficacy; their clinical benefit may diminish over time and side effects may be severe. Adding another therapeutic line may increase the rate of success. BoNT/A was found to be effective in reducing pain intensity and attack frequency in a number of heterogeneous studies. Modalities of administration in trigeminal neuralgia were either intradermal and submucous or around the trigeminal branches. Intradermal administration of 100 UI was performed at the site of pain and the trigger point, 5 UI/point, at a distance of 15 mm between points. If the pain involved the oral mucosa, submucous injections were administered, 2,5 UI/point, in the same manner. After one week, there was complete or adequate pain relief (87%). The main side effect was mild facial asymmetry at the injection site in a small proportion of cases, which resolved spontaneously [32,33].

One injection provided a significant analgesic effect. For patients unresponsive to one injection, repeating the procedure after 2 weeks led to better control of pain, with no increase in the number of side reactions. Regarding the total dose per area, research suggests that low doses (25–40 UI) and high doses (75–100 UI) have similar effects in the short term, whereas high doses have a better effect in the long term, with no additional adverse reactions [34,35].

Injecting the maxillary and mandibular roots with 50 UI botulinum toxin (Botox) offered statistical and clinical improvement in pain intensity and attack frequency, beginning in the first week and persisting for six months. However, about half of the 27 patients experienced recurrence and asked for a second injection [36].

Intradermal injections may be painful, thus local anaesthesia with topical lidocaine or prilocaine was provided; some researchers added inhalatory nitrous oxide 5 min before and throughout the procedure [36].

First-line postherpetic neuralgia treatment includes topical lidocaine, anticonvulsants, and antidepressants. Botulinum toxin has been suggested as a second-line treatment. The tactile allodynia area was mapped and the toxin was injected subcutaneously at sites 10–20 mm apart for a total maximum dose of 100 UI. Pain relief was significant at 7 days and persisted for 4 months [37,38].

Botulinum toxin certainly has a place in postherpetic and trigeminal neuralgia, with some researchers raising the question of whether it should remain a second-line treatment or become a first-line treatment, given its efficacy and tolerability [65].

Peripheral diabetic neuropathy is a symmetrical distal sensory and motor polyneuropathy with a wide spectrum of therapeutic agents, such as antidepressants, anticonvulsants, and opioids. They are reported to lack long-lasting pain relief, and to carry poor tolerability and disturbing side effects. Topical lidocaine and capsaicine (high dose) or botulinum toxin have been studied as alternatives [66].

A total of 100 UI BoNT/A injected intradermally into the dorsal foot in a grid distribution pattern provided analgesia beginning on the first week and maintained for up to three months with no side effects [39,40]. Based on the small number of studies on peripheral diabetic neuropathy with particular localization to the dorsum of the foot, botulinum toxin may act as an adjunctive treatment to first-line modalities.

In allodynia from type I complex regional pain syndrome (CRPS), intradermal and subcutaneous administration of botulinum toxin offered less satisfactory results: a significant percentage of patients reported the procedure to be intolerable and had no pain relief at 3 and 8 weeks [41].

In one case report [42], shoulder CRPS type I secondary to acromioclavicular subluxation with failure of conventional pharmacological and rehabilitative strategies was successfully treated with 100 UI botulinum toxin A administered within the glenohumeral joint. Further research is needed to confirm the results, as studies on this topic are scarce.

Complex regional pain syndrome of the upper limb associated with consecutive dystonia was treated with intramuscular injection of botulinum toxin with electromyographic guidance. The muscles were selected based on patient complaints, hypertrophy, spasm, and tenderness at palpation. Doses varied between 10 and 20 UI per muscle. Results showed analgesic effects and muscle relaxation [43].

Sympathetic blockade has been used for a long time in the treatment of CRPS type I; however, the relief is transitory. Since the pre-ganglionic sympathetic nerve terminals are cholinergic, botulinum toxin was proposed to enhance the blockade. For lower limb CRPS, researchers administered botulinum toxin A (75–100 UI), added as an anaesthetic blockade, to the sympathetic lumbar ganglia to obtain a higher and more durable effect [44]. Some successful attempts were made in the use of botulinum toxin type B in addition to anaesthesia for lumbar sympathetic blockade, with complete resolution of lower limb CRPS. The rationale behind botulinum toxin B use was the possibility of using it in cases of resistance to botulinum toxin A, with a longer blocking effect. There were more adverse effects following BoNT/B administration, including dry mouth and dysphagia; however, these were of low or moderate intensity and disappeared with repeated administration [45,46,47].

Neuropathic pain may benefit from BoNT/A as a second or third line of therapy in cases of recurrent or persistent pain. The main challenge is making the administration tolerable, for example by adding some sort of anaesthesia (either local or general).

### 2.7. Carpal Tunnel Syndrome

Cases of mild and moderate severity were treated with 30 UI botulinum toxin A under sonographic intracanal guidance to obtain a reduction in paraesthesia and nocturnal pain [67].

Another study used intramuscular administration of botulinum toxin into the motor points of flexor digitorum profundis (10 UI), flexor digitorum superficialis (10 UI), and flexor pollicis longus (5 UI) to reduce tension in the tendons travelling to the carpal tunnel and, consequently, the pressure on the median nerve. Results showed significant clinical improvement as well as distal motor latencies and sensor nerve conduction velocity improvements after 12 weeks [48].

Intracanal or intramuscular (flexor muscles) administration of BoNT/A may reduce pressure on the median nerve through the tunnel and improve clinical outcomes.

### 2.8. Morton Neuroma

Patients with Morton’s neuroma who experienced long-lasting pain and failure of conservative therapies, including corticosteroid injections, received 50 UI botulinum toxin A in the area of the neuroma (identified on MRI exam), with 70% of patients reporting pain relief and foot function improvement at one-month and three-month follow-ups. Researchers noticed that the results at 3 months were better than at 1 month, in contrast to the evolution of the antispastic effect, which peaked at 1–3 weeks, was followed by a plateau for 1–2 months, and needed re-injection every 3 months [49].

Refractory cases may require BoNT/A administration, with a possibility of a second administration for resolution of the symptoms.

### 2.9. Myofascial Pain Syndrome

The pathophysiology of myofascial pain syndrome is unclear, as it plays the role of muscular contracture in pain generation. Many therapeutic approaches have been suggested, including local injections with anaesthetics, saline, corticosteroids, or dry-needling—with varying results. Adjunctive physiotherapy is critical for maximizing the pharmacological strategy [50].

For upper back and shoulder myofascial pain syndrome occurring over a long duration (at least 6 months), palpatory identification and injection of botulinum toxin to accessible trigger points (with doses of 10–20–40 UI per point) produced significant analgesic effect and reduced fatigue and work disability at 4 weeks. Smaller doses of 5 UI per trigger point, with a total number of 3–7 trigger points per patient, offered no advantage compared with simple saline injections. Other researchers found equal analgesic effects of botulinum toxin and local anaesthetics injected directly into the trigger points. Thus, treatment should be reserved for patients who fail to respond to other injectable therapies, mainly due to the higher cost of this pharmaceutical [51,52,53,54].

For profound muscles involved in myofascial pain syndrome (i.e., iliopsoas and quadratus lumborum), a single injection under fluoroscopic guidance was found to produce pain relief similar to simple saline or local anaesthetic [68].

When chronic myofascial pain was accompanied by chronic muscle spasm, intramuscular injection of botulinum toxin together with local anaesthetic followed by physiotherapy offered significantly better pain relief and a longer duration of analgesia compared with corticosteroids, which lasted up to three months. Doses varied according to the dimensions of the muscle, for instance, the piriformis muscle received 150 UI, the iliopsoas muscle received 100 UI, and the scalenus anterior received 80 UI [69].

The lack of consensus between studies on the use of botulinum toxin for myofascial pain might be explained by the complex and partially unknown pathophysiologic mechanism, incorrect dosage, or lack of therapeutic effect. Further research is required.

## 3. Conclusions

Chronic or recurrent pain due to a variety of musculoskeletal disorders is a complex syndrome with incompletely known mechanisms and a variety of therapeutic approaches and success rates. In many cases, first-line treatments are well-defined, whereas their failure presents the physician and patient with a large range of therapies with heterogeneous studies and results. Botulinum neurotoxin plays a role in chronic pain management based on its nociceptive neurotransmitter blockage and anti-inflammatory effect. Dosing and timing of BoNT administration in chronic or refractory cases of plantar fasciitis and other tendinopathies, knee osteoarthritis and arthroplasty, and joint contractures and neuropathic pain are subjects of thorough and promising research.

Future directions: As more relevant information on the pain-relieving capabilities of BoNT is gathered, more therapeutic protocols may include this substance as a high-grade line of therapy. Research on doses and timing will provide a scientific basis for personalized therapies.

## Figures and Tables

**Figure 1 biomedicines-11-01888-f001:**
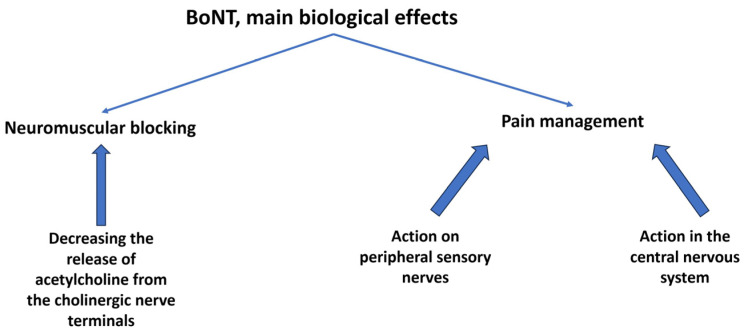
Proposed BoNT mechanisms.

**Table 1 biomedicines-11-01888-t001:** Main BoNT indications of analgesia—(table generated by the author).

Disease	Administration	Number of Studies
Plantar fasciitis	Intrafascial Intramuscular (short plantar muscles) Intramuscular (medial head gastrocnemius)	10
Knee osteoarthritis	Intra-articular	6
Painful knee arthroplasty	Intra-articular Intramuscular (for contractures)	3
Painful muscular contractures	Intramuscular	3
Chronic lateral epicondylitis	Intratendinous Intramuscular	6
Neuropathic pain	Intradermal, subcutaneous/submucosal Perinervous Sympathetic block	16
Carpal tunnel syndrome	Intracannalar Intramuscular	1
Morton neuroma	Perineural	1
Myofascial pain syndrome	Trigger point injection	6

**Table 2 biomedicines-11-01888-t002:** Studies on BoNT and musculoskeletal pain syndromes.

Author	Type of Study	Disease	Methodology of Injection	Timing	Outcomes	Results
Babcock, 2005 [3]	Randomized, double-blind, placebo-controlled, 43 pts	Plantar fasciitis, refractory,	Intrafascial Babcock method (Botox)	3 weeks, 8 weeks	Pain Function (Maryland Foot Score)	Significant improvement versus placebo
Placzek, 2006 [4]	Open case series, 9 pts	Plantar fasciitis, refractory	Intrafascial 200 UI BoNT/A (Dysport)	2, 6, 10, 14 weeks	Pain	Significant improvement
Huang, 2010 [5]	Randomized, double-blind, 50 pts	Plantar fasciitis, refractory	Intrafascial 50 UI BoNT/A (Botox) Ultrasound guidance	3 weeks, 3 months	Pain, Thickness of the plantar fascia, Walking	Reduced pain, Reduced fascia thickness, Walking parameters improved Significantly.
Chou, 2011 [6]	Case report	Plantar fasciitis, refractory	Intrafascial Babcock method	1, 3, 5, and 7 weeks	Pain, Thickness of the plantar fascia (ultrasound)	Pain decreased, Thickness of the fascia decreased significantly.
Peterlein, 2012 [7]	Randomized, double-blind, placebo-controlled, 40 pts	Plantar fasciitis, refractory	Intrafascial 200 UI BoNT/A (Dysport)	2, 6, 10, 14, 18 weeks	Pain	Pain decreased but not significantly compared with placebo
Diaz-Llopis, 2013 [8]	Prospective, observational, 24 pts	Plantar fasciitis	Intrafascial Babcock method	6 months, 12 months	Pain Function (FHSQ)	Significant improvement
Elizondo-Rodriguez (2021) [9]	Randomized, controlled, double-blind, 78 pts	Plantar fasciitis, refractory (BoNT/A, anesthetic, corticosteroid)	Intrafascial 200 UI (Dysport)/anesthetic (5 mL ropivacaine)/1 mL betamethasone	2 weeks, 1, 3, and 6 months	Pain Function Plantar fascia thickness	All groups improved significantly, but no difference between them
Ahmad, 2017 [10]	Prospective, randomized, placebo-controlled, 89 pts	Plantar fasciitis, refractory	Intramuscular Flexor digitorum brevis 100 UI inobotulinum (Xeomin)	6, 12 weeks, and 6, 12 months	Pain, Function (FAAM)	Pain reduction and Functional improvement Significant
Radovic, 2020 [11]	Case series, 4 pts	Plantar fasciitis, refractory	Intramuscular Abductor hallucis 50 UI Quadratus plantae 50 UI (Dysport)	1, 3, 6, 12, 26 weeks	Pain Function (FAAM)	Pain reduction, Functional improvement
Abbasian, 2019 [12]	Prospective, randomized, placebo-controlled, double-blind, 32 pts	Plantar fasciitis, refractory	Intramuscular Medial gastrocnemius 70 UI	1, 3, 6, and 12 months	Pain Function	Pain reduction, Functional improvement Significant
Boon, 2010 [13]	Prospective, randomized, double-blind, 60 pts	Knee osteoarthritis grade II, III, (CS, low dose BoNT/A, high-dose BoNT/A)	Intraarticular Low-dose 100 UI High-dose 200 UI	8 weeks, 6 months	Pain Function (WOMAC)	Pain improved significantly with low dose and CS
Chou, 2010 [14]	Prospective, observational, 24 pts	Knee osteoarthritis grade III, IV	Intraarticular 100 UI BoNT/A (Botox), 2 injections 3 months apart	3, 6 months	Pain Function (WOMAC)	Pain relief, Functional improvement
Hsieh, 2016 [15]	Prospective, randomized, controlled, 46 pts	Knee osteoarthritis grade II, III	Intraarticular 100 UI BoNT/A	One week, 6 months	Pain Function (WOMAC)	Pain relief, Functional improvement in BoNT/A group versus control
Najafi, 2019 [16]	Prospective, single group, 46 pts	Knee osteoarthritis, grade III, IV	Intraarticular 250 UI BoNT/A (Dysport)	4 weeks	Pain Function (KOOS)	Pain relief, Functional improvement
Mendes, 2019 [17]	Prospective, randomized, controlled, 105 pts	Knee osteoarthritis, grade II, III, 3 groups (BoNT/A 100 UI, 200 UI or CS)	Intraarticular 100 UI BoNT/A (Botox), 40 mg triamcinolone, saline	4 weeks, 12 weeks	Pain Function (WOMAC)	No benefits in the BoNT/A group, CS group improved pain and function
Bao, 2018 [18]	Prospective, single-blinded, randomized, controlled, 60 pts	Knee osteoarthritis, grade II, III, IV 3 groups (HA, BoNT/A, saline) + exercise	Intraarticular 100 UI BoNT/A (Botox)	8 weeks	Pain Function (WOMAC) Quality of life	HA and BoNT/A followed by exercise effective in reducing pain and disability. BoNT/A more effective
Singh, 2010 [19]	Randomized, placebo-controlled, 54 pts	Painful knee arthroplasty	Intraarticular 100 UI BoNT/A (Botox)	2 months	Pain Function (WOMAC)	Pain reduction, Functional improvement significant
Singh, 2014 [20]	Retrospective, 11 pts	Painful knee arthroplasty	Intraarticular 100 UI BoNT/A (Botox)	28 months	Pain Function (WOMAC)	Pain reduction, Functional improvement
Smith, 2016 [21]	Randomized, double-blind, placebo-controlled, 14 pts	Flexion contracture knee arthroplasty	Intramuscular (hamstrings) 100 UI BoNT/A (Botox)	1, 6, 12 months	Extension ROM	Extension improved significantly in the BoNT/A group
Vahedi, 2020 [22]	Retrospective, 19 pts	Flexion contracture knee arthroplasty	Intramuscular (hamstrings, gastrocnemius)	31 months	Knee extension	Extension improved
Daffunchio, 2016 [23]	Prospective, 17 pts	Flexion contracture, hemophilia	Intramuscular (calf and hamstrings muscles, fascia lata) 100 UI BoNT/A	1, 3, 6, and 12 months	Knee ROM	ROM increased significantly
Bhave, 2009 [24]	Case series, 10 pts	Hip flexion contracture, arthroplasty	Intramuscular (adductor magnus, brevis, tensor fascia lata, rectus femoris) 100 UI BoNT/A	12 months	Hip ROM Function (Harris hip scores)	ROM increased Functional improvement
Wong, 2005 [25]	Randomized, double-blind, placebo-controlled, 60 pts	Chronic lateral epicondylitis	Intramuscular 60 UI BoNT/A (Dysport)	4, 12 weeks	Pain Function (grip strength)	Pain relief significant
Placzel, 2007 [26]	Randomized, placebo-controlled, double-blind, 130 pts	Chronic lateral epicondylitis	Intramuscular 60 UI BoNT/A (Dysport)	2, 6, 12, 18 weeks	Pain Function (grip strength)	Pain relief, Functional improvement significant
Hayton, 2005 [27]	Randomized, placebo-controlled, double-blind, 40 pts	Chronic lateral epicondylitis	Intramuscular 50 UI BoNT/A (Botox)	3 months	Pain Function (grip strength)	No significant difference between groups
Espandar, 2010 [28]	Randomized, placebo-controlled, double-blind, 48 pts	Chronic lateral epicondylitis	Intramuscular 60 UI BoNT/A	4, 8, 16 weeks	Pain Function (grip strength)	Pain relief, Functional improvement significant
Creuze, 2018, [29]	Randomized, placebo-controlled, double-blind, 67 pts	Chronic lateral epicondylitis	Intramuscular (ECRB) 40 UI BoNT/A (Dysport)	1, 3 months	Pain	Pain reduction significant
Cogne, 2018, [30]	Open study, prospective, 50 pts	Chronic lateral epicondylitis	Intramuscular (ECRB) 40 UI BoNT/A (Dysport)	3, 6, 9, 12 months	Pain	After 2 injections, 90% pts reported significant pain reduction
Ranoux, 2008 [31]	Randomized, placebo-controlled, double-blind, 29 pts	Postherpetic and posttraumatic neuralgia	Intradermally 100 UI BoNT/A (Botox)	4, 12, 24 weeks	Pain Allodynia	Long-lasting and significant reduction in pain
Wu, 2019 [32]	Retrospective cohort, 104 pts	Trigeminal neuralgia	Intradermally, 100 UI BoNT/A	Every 2 months, up to 12 months	Pain	Pain reduction significant
Zuniga, 2008 [33]	Case series, 12 pts	Trigeminal neuralgia	Subcutaneous, 20–50 UI BoNT/A (Botox)	8 weeks	Pain	Pain reduction significant
Zhang, 2014 [34]	Randomized, placebo-controlled, double-blind, 84 pts	Trigeminal neuralgia	Intradermally, 25 or 75 UI BoNT/A	8 weeks	Pain	Pain reduction significant, Both doses have similar efficacies
Zhang, 2019, [35]	Retrospective, 152 pts	Trigeminal neuralgia	Intradermally, 50, 70, and over 70 UI BoNT/A	Up to 28 months	Pain	Pain reduction at 2 weeks, persistent at 28 months
Turk Boru, 2017 [36]	Prospective, observational, 27 pts	Trigeminal neuralgia	Perineural 50 UI BoNT/A per root (Botox)	2, 6 months	Pain	Pain and attack frequency reduction significant
Xiao, 2010 [37]	Prospective, randomized, placebo-controlled, double-blind, 60 pts	Postherpetic neuralgia	Subcutaneous 100 UI BoNT/A versus lidocaine versus saline	3 months	Pain, allodynia	Significant pain reduction at 7 days, persistent at 3 months
Apalla, 2013 [38]	Prospective, randomized, placebo-controlled, double-blind, 30 pts	Postherpetic neuralgia	Subcutaneous 100 UI BoNT/A	2, 4, 6, 8, 10, 12 weeks	Pain, Sleep quality	Pain reduction, Sleep improvement significant
Ghasemi, 2014 [39]	Prospective, randomized, placebo-controlled, double-blind, 40 pts	Diabetic neuropathy in lower limbs	Intradermal 100 UI BoNT/A (Botox)	3 weeks	Neuropathy pain scale	Improvement of all sensations except cold
Yuan, 2009 [40]	Prospective, randomized, placebo-controlled, double-blind, 18 pts	Diabetic neuropathy in lower limbs	Intradermal 100 UI BoNT/A (Botox)	4, 8, 12 weeks	Pain	Significant pain reduction at all time points
Safarpour, 2010 [41]	Pilot study, 14 pts	Complex regional pain syndrome	Indermal and subcutaneous, 40–200 UI BoNT/A	3 weeks, 2 months	Pain	Failure of pain reduction
Bellon, 2019 [42]	Case report	Complex regional pain syndrome, upper arm	Intraarticular 100 UI BoNT/A (Xeomin)	1 and 4 months	Pain	Pain reduction
Kharkar, 2011 [43]	Case series, 37 pts	Complex regional pain syndrome upper girdle	Intramuscular 100 UI BoNT/A	4 weeks	Pain	Pain reduction
Carroll, 2009 [44]	Prospective, randomized, double-blind, pilot, 9 pts	Complex regional pain syndrome, lower limb	Lumbar sympathetic block 75 UI BoNT/A	1 month	Pain	Pain reduction
Choi, 2015 [45]	Case report, 2 pts	Complex regional pain syndrome, lower limb	Lumbar sympathetic block 5000 UI BoNT/B	1 month	Pain, Skin color	Pain reduction, skin color normal
Lee, 2018 [46]	Retrospective, comparative, 18 pts	Complex regional pain syndrome, lower limbs	Lumbar sympathetic block, 100 UI BoNT/A versus 5000 UI BoNT/B	1 week	Pain	Both groups had pain relief
Tsai, 2006 [47]	Prospective, open-label, pilot study, 5 pts	Carpal tunnel syndrome	Intracannalar 60 UI BoNT/A	3 months	Pain, electrophysiological studies	Pain relief, electrophysiological parameters improvement
Climent, 2013 [48]	Prospective, pilot study, 17 pts	Unresponsive Morton neuroma	Perineural, 50 UI BoNT/A	3 months	Pain Function (FHSQ)	Pain relief at rest and walking, at all time points
Ojala, 2006 [49]	Prospective, double-blind, placebo-controlled, 31 pts	Myofascial syndrome	Trigger point injection, a total of 15–50 UI BoNT/A/session, 2 sessions 4 weeks apart	4, 8 weeks	Pain	No difference between groups
Kamanli, 2005 [50]	Prospective, single-blind, 87 pts	Myofascial syndrome	Trigger point injection, 10–20 UI versus lidocaine versus dry needling	1 month	Pain Subjective complaints	No statistical difference between groups
Gobel, 2006 [51]	Randomized, double-blind, placebo-controlled, 145 pts	Myofascial syndrome	Trigger point injection, a total of 40 UI BoNT/A (Dysport)	12 weeks	Pain	Pain relief significant
Graboski, 2005 [52]	Randomized, double-blind, 18 pts	Myofascial syndrome	Trigger point injection, 25 UI BoNT/A per point, maximum 8 points/pt versus bupivacaine	10 weeks	Pain	Both groups improved significantly, without a difference
De Andres, 2010 [53]	Prospective, randomized, double-blind, controlled, 27 pts	Myofascial syndrome iliopsoas and/or quadratus lumborum	Intramuscular 50 UI BoNT/A/muscle versus bupivacaina	15, 30, 90 days	Pain	Both groups improved significantly, without a difference
Porta, 2000 [54]	Randomized, comparative, placebo-controlled, double-blind, 40 pts	Myofascial syndrome	Intramuscular 80–150 UI/muscle BoNT/A versus CS	30, 60 days	Pain Spasms	BoNT group improved pain and spasms significantly

FHSQ, Foot Health Status Questionnaire; FAAM, Foot and Ankle Ability Measures; WOMAC, Western Ontario and McMaster Universities Osteoarthritis Index; CS, corticosteroid; KOOS, Knee injury and Osteoarthritis Outcome Score; HA, hyaluronic acid; ROM, range of motion; ECRB, extensor carpi radialis brevis.

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
