# Peer review of "Pain Modulation in Chronic Musculoskeletal Disorders: Botulinum Toxin, a Descriptive Analysis"

_biomedicines, 2023, doi:10.3390/biomedicines11071888_

Round 1

Reviewer 1 Report

Thank you for the opportunity to review your manuscript, “Pain modulation in chronic musculoskeletal disorders - botulinum toxin”.

The type of study should be included in the title to make it clear to readers what the article is about.

Line 39. It should be consistent with the citation format.

The study's design, a literature review, should be included.

Reading the article, I understand why the authors aim to highlight some painful musculoskeletal pathologies since the authors present a series of pathologies where they consider that botulinum toxin may be susceptible to be used and describe the evidence that there is.

The wording of the article reads more like a loose note draft than a review article. 

In the general section on tendinopathies, evidence is only shown for lateral epicondylalgia.

There needs to be a general discussion of the subject.

Author Response

Reply to Reviewer 1.

Thank you for your comments, I hope I would fulfil them in order in increase the quality of our paper.

The type of study should be included in the title to make it clear to readers what the article is about.

Reply: We changed the title accordingly.

Line 39. It should be consistent with the citation format.

Reply: We made the change.

The study's design, a literature review, should be included.

Reply: we added the appropriate issue.

Reading the article, I understand why the authors aim to highlight some painful musculoskeletal pathologies since the authors present a series of pathologies where they consider that botulinum toxin may be susceptible to be used and describe the evidence that there is.

Reply: Yes, this was the aim. As there are a number of therapeutic failures, botulinum toxin may be a choice.

The wording of the article reads more like a loose note draft than a review article.

Reply: we asked for adequate English phrasing. 

In the general section on tendinopathies, evidence is only shown for lateral epicondylalgia.

Reply: as lateral epicondylitis is the main topic in this section, we changed the title.

There needs to be a general discussion of the subject.

Reply: we addressed the issue.

Reviewer 2 Report

This review summarises the clinical evidence in the literature regarding the effects of Botulinum neurotoxin on different musculoskeletal diseases characterised by chronic pain. The authors provide the main findings for the use of BoTN and suggest what could be the future directions for a wider use of BoTN as a biopharmaceutical.

The manuscript is well written and includes an up-to-date bibliography. Some points can be addressed aiming to improve the quality of the work.

1.       A careful editing would be useful to correct some errors, e.g. is reveals in the Abstract.

2.       A summative table including the main findings of the citations for the different diseases could make the text easier to follow. As it stands, Table 1 is not very useful containing minimal information.

3.       An graphical illustration describing the mechanism of action of BoTN and the different effects for the diseases described in the text would add to the manuscript.

A careful editing would be useful to correct some errors, e.g. is reveals in the Abstract.

Author Response

Reply to Reviewer 2:

Thank you for your comments, we hope the paper would finally improve.

This review summarises the clinical evidence in the literature regarding the effects of Botulinum neurotoxin on different musculoskeletal diseases characterised by chronic pain. The authors provide the main findings for the use of BoTN and suggest what could be the future directions for a wider use of BoTN as a biopharmaceutical.

The manuscript is well written and includes an up-to-date bibliography. Some points can be addressed aiming to improve the quality of the work.

  1. A careful editing would be useful to correct some errors, e.g. is reveals in the Abstract.

Reply: we asked for professional help.

  1. A summative table including the main findings of the citations for the different diseases could make the text easier to follow. As it stands, Table 1 is not very useful containing minimal information.

Reply: we improved table 1 and added table 2 to show the main papers on this topic

  1. An graphical illustration describing the mechanism of action of BoTN and the different effects for the diseases described in the text would add to the manuscript.

Reply: we inserted such an illustration

Reviewer 3 Report

This study is a review of botulinum-toxin use in different diseases. This study is generally interesting, but in my opinion is to much descriptive without any clinical application. There is a lack of summary or some conclusion after each paragraph, therefore it is difficult to apply the information contained therein in practice. The lack of a quantitative comparison of the results of available studies means that the descriptions mean nothing. This study does not meet scientific quality. 

Author Response

Reply to Reviewer 3.

Thank you for your comments. We modified the manuscript to increase her value and to fulfill the exigencies.

This study is a review of botulinum-toxin use in different diseases. This study is generally interesting, but in my opinion is to much descriptive without any clinical application. There is a lack of summary or some conclusion after each paragraph, therefore it is difficult to apply the information contained therein in practice. The lack of a quantitative comparison of the results of available studies means that the descriptions mean nothing. This study does not meet scientific quality. 

Round 2

Reviewer 1 Report

The authors have responded to all my comments. I believe that with the new contributions, the article may be of interest.

Author Response

Thank you for your interest.  

Reviewer 2 Report

I do not see the figure and the supplemental table should be in the main manuscript

Author Response

Thank you for your comment.

We introduced the figure and Table 2 into the main document.

Reviewer 3 Report

The authors have improved some parts of this manuscript, but still my comments did not addressed. The manuscript is to much descriptive without any clinical application. There is a lack of conclusions or any key points throghout the paragraph. I support my previous decision, that it does not meet the scientific standards. 

Author Response

Thank you for your comment.

Indeed, the paper has a descriptive aspect. Research on pain management with the aid of BoNT is scarce. We searched for directions and belived that, in certain refractory or recurrent cases, physicians may be helped by new information. A more comprehensive study would be possible with increasing interest and number of published papers. We added conclusion or discussion at the end of every section. 

Round 3

Reviewer 2 Report

The authors sufficiently addressed my comments.

Reviewer 3 Report

In my opinion, the work is still poor despite the corrections. Adding one sentence at the end of each chapter doesn't change much. I leave the further fate of this work to the editor's decision